# Oncolytic Virotherapy: The Cancer Cell Side

**DOI:** 10.3390/cancers13050939

**Published:** 2021-02-24

**Authors:** Marcelo Ehrlich, Eran Bacharach

**Affiliations:** Shmunis School of Biomedicine and Cancer Research, George S. Wise Faculty of Life Sciences, Tel Aviv University, Tel Aviv 6997801, Israel

**Keywords:** oncolytic viruses, immunoediting, oncogenic signaling, RAS, DNA methyltransferase inhibitor (DNMTi), viral mimicry, epigenetic silencing

## Abstract

**Simple Summary:**

Oncolytic viruses (OVs) are a promising immunotherapy that specifically target and kill cancer cells and stimulate anti-tumor immunity. While different OVs are endowed with distinct features, which enhance their specificity towards tumor cells; attributes of the cancer cell also critically contribute to this specificity. Such features comprise defects in innate immunity, including antiviral responses, and the metabolic reprogramming of the malignant cell. The tumorigenic features which support OV replication can be intrinsic to the transformation process (e.g., a direct consequence of the activity of a given oncogene), or acquired in the course of tumor immunoediting—the selection process applied by antitumor immunity. Oncogene-induced epigenetic silencing plays an important role in negative regulation of immunostimulatory antiviral responses in the cancer cells. Reversal of such silencing may also provide a strong immunostimulant in the form of viral mimicry by activation of endogenous retroelements. Here we review features of the cancer cell that support viral replication, tumor immunoediting and the connection between oncogenic signaling, DNA methylation and viral oncolysis. As such, this review concentrates on the malignant cell, while detailed description of different OVs can be found in the accompanied reviews of this issue.

**Abstract:**

Cell autonomous immunity genes mediate the multiple stages of anti-viral defenses, including recognition of invading pathogens, inhibition of viral replication, reprogramming of cellular metabolism, programmed-cell-death, paracrine induction of antiviral state, and activation of immunostimulatory inflammation. In tumor development and/or immunotherapy settings, selective pressure applied by the immune system results in tumor immunoediting, a reduction in the immunostimulatory potential of the cancer cell. This editing process comprises the reduced expression and/or function of cell autonomous immunity genes, allowing for immune-evasion of the tumor while concomitantly attenuating anti-viral defenses. Combined with the oncogene-enhanced anabolic nature of cancer-cell metabolism, this attenuation of antiviral defenses contributes to viral replication and to the selectivity of oncolytic viruses (OVs) towards malignant cells. Here, we review the manners by which oncogene-mediated transformation and tumor immunoediting combine to alter the intracellular milieu of tumor cells, for the benefit of OV replication. We also explore the functional connection between oncogenic signaling and epigenetic silencing, and the way by which restriction of such silencing results in immune activation. Together, the picture that emerges is one in which OVs and epigenetic modifiers are part of a growing therapeutic toolbox that employs activation of anti-tumor immunity for cancer therapy.

## 1. Introduction

The present review focuses on the differential and enhanced susceptibility of cancer cells to oncolytic viruses (OVs). We propose that such hyper-susceptibility of the malignant cells stems from unique features of the cancer-cell milieu, including defective antiviral responses and metabolic reprograming. The sources of such tumor-cell specific alterations comprise a combination of factors, which are intrinsic to the tumor cell—e.g., oncogene-stimulated signaling, and/or extrinsic ones; e.g., selective pressure applied by the tumor immune microenvironment. We begin by focusing on the cancer-cell *per se*, analyzing how oncogene-induced modifications serve to optimize the intracellular environment towards OV replication. To this end, we employ RAS-activated pathways as a pivot, exemplifying how this intrinsic oncogenic pathway modulates antiviral responses. We then proceed to focus on the immunoediting of tumors, as this provides a critical extrinsic (selective) source of alterations to cancer-cell autonomous immune functions. Given the overlap in the immune-activation-potential of a cancer cell and its ability to raise antiviral responses, the selective pressure applied by anti-tumor immunity results in both decreased immunogenicity and in defective antiviral responses. We finalize our review by focusing on oncogene-stimulated DNA methylation in the context of immune evasion, as an example of how the two processes (oncogenic signaling and immunoediting) converge to influence OVs-cancer-cell interactions. Our focus on DNA methylation stems from its prominence as a molecular mechanism for silencing of cell-autonomous immune responses. In this context, we also discuss the reversal of this form of epigenetic silencing, which may elicit tumor immunogenicity through the expression of endogenous retroelements, thus generating a “viral mimicry” state, emulating the immune-stimulatory potential of OVs.

## 2. Defects to Cell Autonomous Immunity and Metabolic Reprogramming Optimize the Cancer Cell Milieu towards Viral Infection

### 2.1. Cell Autonomous Immunity: The Antiviral Response

The cell autonomous immune response provides the first line of defense against cellular pathogens, including viruses [1]. To deal with a wide variety of pathogens, activation of cell autonomous immunity occurs in an antigen-independent fashion. Instead, it relies on the ability of the cell to recognize molecular patterns which are abundant in pathogens (pathogen-associated-molecular patterns, PAMPs), yet relatively absent in healthy cells. These molecular patterns are recognized by pattern recognition receptors (PRRs), which survey distinct cellular compartments for the presence of PAMPs. In addition, aberrant intracellular localization of nucleic acids (e.g., intra-endosomal localization of RNA or DNA, or cytoplasmic localization of DNA) also serves to discern between nucleic acids of cellular vs. pathogen origin, and when detected, stimulates cell autonomous immune responses (reviewed in [2,3,4]). A prototypic PAMP is double stranded RNA (dsRNA), an obligatory molecular pattern of viral infection, which may be recognized by toll-like receptor 3 (TLR3) upon exposure to the endosomal lumen, or by RNA helicases—the retinoic acid-inducible gene I (RIG-I) and the melanoma differentiation-associated gene 5 (MDA5) upon exposure in the cytoplasm [1,5]. DNA too can serve as a PAMP, depending on its composition or intracellular localization. In these contexts, TLR9 recognizes DNA molecules rich in unmethylated CpG sequences, as commonly occurs in genomes of viruses and bacteria [6]; while cytoplasm-localized DNA is recognized by cyclic GMP-AMP Synthase (cGAS) [7]. In a typical case, exemplified here by the cellular response to RNA virus infections, PAMP-induced PRR signals are transduced through mitochondrial antiviral-signaling protein (MAVS), Tank-binding kinase 1 (TBK1) and IKKs; resulting in the activation of nuclear factor kappa-light-chain-enhancer of activated B cells (NF-kB) and interferon (IFN)-regulatory factors (IRFs) 3 and 7. These in turn translocate to the nucleus and mediate the transcriptional activation of type I or type III IFNs (e.g., IFN-β). Following synthesis and secretion, IFNs activate Janus kinase (JAK)- signal transducer and activator of transcription (STAT) signaling, resulting in STAT-mediated massive amplification of the cell autonomous immune response via the induction of IFN-stimulated-genes (ISGs) [1,5,8].

### 2.2. Oncogene-Induced Perturbations to Antiviral Responses: A Reduction in Impediments to Viral Replication

Oncogene-induced perturbations to antiviral responses are prominent molecular mechanisms by which the cancer-cell milieu becomes optimized towards OV replication. To exemplify this concept, we focus on such effects related to oncogenic RAS. Oncogenic mutations in RAS, a GTP-activated molecular switch, ensue exposure to genotoxic agents, and are estimated to occur in 16–30% of all human cancers, with highest incidence in pancreatic (90%) and colon (50%) cancers; and considerable portions of melanoma and lung adenocarcinoma [9,10,11]. Activated RAS (either because of oncogenic mutations or following stimulation of upstream growth receptors) stimulates downstream signaling pathways mediated by phosphatidylinositol 3 (OH)-kinase (PI3K), RAL guanine nucleotide dissociation stimulator (RALGDS) family members, and members of the RAF family, which activate the RAF/MEK/ERK pathway [12]. Thus, RAS functions as a multi-pronged signaling node, which upon activation, endows tumor cells with multiple malignancy-associated features. Multiple lines of evidence place RAS, and its associated signaling pathways, as negative regulators cell autonomous immunity. 

#### 2.2.1. RAS-Mediated Regulation of Immune Transcription Factors

In accord with oncogene-mediated regulation of gene expression programs, a critical mechanism by which they modify immune/antiviral functions of tumor cells is through regulation of the expression of immunity-related transcription factors. In HRAS transformed murine fibroblasts, and RAS-transformed human cancer cells, MEK-ERK signaling was shown to negatively regulate IRF-1-dependent transcription of IRF1 and STAT2 [13,14], thus hampering IFN responses, and supporting the replication of oncolytic vesicular stomatitis virus (VSV). In addition to immune-related functions (e.g., as antiviral gene, master regulator of acute inflammation, and main effector of IFNγ signaling), IRF1 was also characterized as a tumor-suppressor [15,16,17,18,19]. Thus, IRF1 inhibition by RAS-MEK is predicted to concomitantly promote tumorigenicity, alter the interactions between tumor- and immune cells and enhance the susceptibility of cancer cells to OVs. Of note, the antagonism of IRF1 function by mitogenic pathways is not restricted to cancer settings. For example, in airway epithelial cells, influenza A virus (IAV) and rhinovirus activate the epidermal growth factor receptor (EGFR, [20])—an upstream activator of the RAS/RAF/MEK/ERK pathway [21]. Activated EGFR diminishes both IRF1 expression and induction of IFN-λ production, thus increasing viral infection. Oncogenic KRAS was shown to inhibit the expression of STAT1, STAT2 and IRF9 (members of the ISGF3 transcription-promoting complex); thus, hampering the basal and IFN-induced expression of ISGs in colorectal cancer cell lines [22]. This effect was proposed to be mediated (at least in part) through the PI3K pathway. Moreover, a recent study employing a murine model of colorectal cancer combining oncogenic KRAS expression with conditional null alleles of adenomatous polyposis coli (*APC*) and *TRP53*, identified repression of IRF2 as a key mechanism for KRAS-induced immune-suppression in colorectal cancer [23]. It should be noted that the roles of IRF2 in cancer are controversial. Thus, while IRF2 expression is downregulated in many different tumor types [24] suggesting potential tumor suppressor roles, other studies proposed pro-tumorigenic functions for IRF2, including via antagonism of IRF1 functions [15,25]. Similarly, while IRF2 was proposed to antagonize IRF1 antiviral responses [26], more recent studies suggest complementary roles for IRF1 and IRF2 in IFN-induced gene expression.

#### 2.2.2. Inhibition of PKR Licenses Cells for Viral Infection

A major antiviral signaling node, which is targeted by RAS-induced signaling, is the dsRNA-activated protein kinase, PKR, which following the binding of dsRNA inhibits protein synthesis via phosphorylation of the eukaryotic initiation factor 2 α (eIF2α) [27,28]. In accord with the enhanced protein synthesis requirements of cancer cells, PKR has been identified as a tumor suppressor in different malignancy settings [28,29,30]; inducing apoptosis upon its activation [31,32]. The notion of PKR as a main antiviral gene is underscored by the numerous inhibitory mechanisms against PKR which are encoded/induced by different viruses [33,34,35,36,37], and by the enhancement of viral replication and viral-induced lethality in PKR-null cells and mice, respectively [38]. Based on this dual role of tumor suppressor and antiviral effector, oncogene-mediated targeting of PKR in general, and its inhibition by the RAS/RAF/MEK/ERK pathway in particular, can be exploited by OVs. For example, wild-type IAV counters PKR via its NS1 protein [39], and via activation of mitogen-activated protein kinase-activated protein kinases (MAPKAPKs) MK2 and MK3 [40]. In accord with PKR being an ISG [38], mutant IAV lacking NS1 replicate only in interferon-deficient systems [41] and perturbation of expression of MK2 or MK3 reduces IAV titers, and enhances PKR activation and eIF2α phosphorylation by the dsRNA mimic polyI:C [40]. In accord with RAF/MEK/ERK-mediated licensing of cells towards IAV infection, IAV shows a strong tropism towards cells expressing active RAF both in vitro and in vivo [42]. Similarly, expression of oncogenic NRAS in melanoma cells, suffices to make them selectively susceptible to oncolysis by IAV lacking NS1 [43]. The centrality of PKR inhibition by the RAS/RAF/MEK/ERK signaling axis in determining susceptibility of cancer cells to OVs is further exemplified by: (i) the requirements of herpes simplex virus 1 (HSV1) Δγ(1)34.5 mutants for MEK-mediated PKR inhibition [44], (ii) the oncotropism of VAI mutant adenovirus towards cells in which RAS inactivates PKR [45], and (iii) the selectivity of the mammalian reovirus towards RAS-transformed cells, which was initially identified as dependent on PKR inactivation [46,47]. This latter tropism has been further dissected and was shown to involve additional mechanisms, including: activation of RAL-GTP exchange factor (RAL-GEF) and the p38 kinase, downstream of RAS [48]; the RAS-mediated enhancement of multiple reovirus infection features including uncoating, particle infectivity, and apoptosis-dependent virion release [49]; and the RAS-mediated inhibition of RIG-I expression/function [50]. In line with the latter inhibitory mechanism, RAF/MEK/ERK activation also hampers RIG-I- and IFN-mediated restriction of VSV replication [51].

#### 2.2.3. Inhibition of Antiviral Responses by RAS-Regulated Factors

Oncogenic RAS may also regulate OV replication through effects on additional oncogenes. For example, the enhanced replication of oncolytic Newcastle disease virus (NDV) depends on RAC1 in highly-malignant RAS-transformed keratinocytes [52]; and RAC1 is a downstream effector of oncogenic RAS [53,54]. In addition, the CDC25 phosphatase, a RAF-regulated oncogene [55], negatively regulates TBK1 through dephosphorylation, inhibiting RIG-I-mediated induction of IFN [56]. Moreover, while oncogenic KRAS increases PKC-βII expression in a murine colon-cancer model [57], this enzyme phosphorylates and inhibits RIG-I, and enhances VSV replication in different cellular settings [58]. The notion of a functional interaction between MDA-5 and oncogenic-RAS is exemplified by the suppression of pro-apoptotic effects of MDA-5 overexpression by either oncogenic RAS or RAF [59]. An additional mode of action is observed for the MYC oncogene, which functions as a crucial effector of oncogenic KRAS, [60,61] and represses, together with the transcriptional repressor MIZ, the type I IFN pathway [61]. Interestingly, inactivation of the tumor suppressor phosphatase and tensin homologue (PTEN), which among its well-documented malignancy-promoting activities [62] accelerates tumorigenesis induced by KRAS [63], results in increased phosphorylation of Ser97 in IRF3, in the negative regulation of IRF-mediated IFN induction upon viral challenge, and in increased viral (VSV) replication [64]. 

Together, the above-mentioned examples (Section 2.2) demonstrate the ability of oncogenic signaling to interfere with all steps of the antiviral response continuum, including PRR-mediated PAMP recognition, IFN induction, JAK/STAT signaling and ISG expression. 

### 2.3. Oncogene-Mediated Stimulation of Anabolism: Supplying the Metabolic Needs of Replicating Viruses

Both viral replication and tumor-cell growth are anabolic processes, i.e., dependent on the biosynthesis of macromolecules (nucleic acids, proteins, lipids and oligosaccharides). As such, both oncogenic transformation and viral infection optimize the cell’s metabolic regulation towards their anabolic needs. The efficiency and extent by which oncogene-induced processes carry out such reprograming is predicted to support enhanced replication of OVs. For example, oncogenic KRAS stimulates anabolic metabolism to maintain pancreatic tumors through activation of MAPK and MYC pathways and the ensuing increased expression of genes which regulate sterol biosynthesis, pyrimidine metabolism and glycosylation [65]. Such metabolically reprogramed cells are characterized by increased glycolytic flux (Warburg effect, [66]) and by glutamine serving as a major carbon source for the tri-carboxylic acid (TCA) cycle [67]. Multiple lines of evidence support the notion that viruses benefit from analogous metabolic reprograming, as different viruses manipulate cell metabolism towards aerobic glycolysis (reviewed in [68,69]) and reprogram glutamine catabolism to optimize virus replication [70]. Similarly, fatty acid synthase (FASN), which regulates the production of long-chain fatty acids [71], is overexpressed in different tumors [71,72], and induced upon oncogenic-RAS-mediated cell transformation [73,74]. Analogous to its role in tumorigenesis, FASN-mediated lipogenesis is required for infection with diverse viruses [75,76,77,78,79]. The similitude of the metabolic requirements of KRAS-transformed tumors and viruses is further exemplified by the effects of inhibitors of dihydroorotate dehydrogenase (DHODH), which perturb *de novo* pyrimidine biosynthesis, selectively inhibit the growth of KRAS mutant cell lines [80] and exhibit broad-range antiviral activity against RNA viruses [81]. 

The multiple effects of oncogenic RAS, which promote viral replication and reduce tumor-cell immunogenicity are schematically depicted in Figure 1.

## 3. Immunoediting Selects for Cancer Cells with Defects in Immune-Stimulatory Abilities

Immunosurveillance and tumor immunoediting are complementary and consecutive processes involving the interaction of a competent immune system with developing tumors. The former refers to the continuous recognition and targeting of malignant cells as a result immune activity. Contrastingly, immunoediting results in the selection of tumor cells with reduced immunogenicity as consequence of selective pressures applied by innate and adaptive immunity. Tumor immunoediting is commonly divided into three phases (the “three E’s”): (i) elimination, where cancer cells are destroyed by immunosurveillance mechanisms; (ii) equilibrium, where cells surviving the initial immune onslaught undergo consecutive rounds of functional, epigenetic and genetic changes. These result in adaptation, i.e., improved fitness of the malignant cells within the tumor microenvironment (TME) co-populated by immune cells; (iii) escape, where outgrowth of resistant clones induces and supports an immunosuppressive microenvironment (reviewed in [82,83], schematically depicted in Figure 2).

### 3.1. Molecular Mechanisms of Immunoediting: Optimization of the Cancer Cell towards Viral Oncolysis

The molecular mechanisms underpinning immunoediting are multifold and include: (i) Increased ability of cancer cells to survive immune-cell-induced death. This occurs through multiple mechanisms including: inactivating mutations, epigenetic silencing or sequestration of components of cell death pathways induced by immune cells [84,85,86,87,88,89,90], overexpression of decoy receptors (reviewed in [91]), or interference with the cancer-cell apoptotic machineries [92]. While, in theory, such interference may make it more difficult for OVs to kill cancer cells by apoptosis, it may also allow for an extension of the period during which the virus replicates, increasing thus the viral titer within the tumor. Of note, OVs have been shown to kill cancer cells via multiple pathways (in addition to apoptosis), including necrosis, necroptosis, pyroptosis, and autophagic cell death (reviewed in [93]), suggesting their ability to circumvent the enhanced resistance to apoptosis of cancer cells. (ii) Reduced immunogenicity of cancer cells. A main mode of loss of immunogenicity are acquired defects to the expression and/or function of the cell’s antigen processing and presentation machineries [94]. This occurs via a broad range of processes including inactivating mutations or epigenetic silencing of MHC-I *per se* or of co-factors required for its expression [95,96,97]; inhibition of signaling pathways that promote MHC-I expression [98,99,100]; or activation of pathways that inhibit MHC-I expression [101,102,103]). Additionally, cancer cells also decrease expression of pro-inflammatory cytokines, such as in the epigenetic silencing of IFN-γ or IFN-κ in cervical cancer and Human Papillomavirus Type 16 (HPV-16)-positive cells, respectively [104,105]; or the reduced expression of pro-inflammatory cytokines in non-small cell lung cancers (NSCLC) [106]. The overlap in the genetic/signaling programs which mediate MHC-I expression, inflammation and antiviral responses, suggests that the downregulation of the former programs in the context of immunoediting should diminish cancer-cell resistance to OV infection. For immune evasion, the reduction in immune stimuli is complemented through increased expression of negative regulators of immune cell function (e.g., programmed cell death-ligand 1 (PD-L1) [107,108,109,110]. In accord with its function as an effector of negative feedback of inflammatory responses, PD-L1 expression is stimulated by IFN-γ, JAK/STAT signaling, and IRF1 [111]; and by TNFα and NF-κB [112]. Given that these pathways mediate cell autonomous immunity, this would suggest that PD-L1 upregulation can be associated with increased resistance to OV infection. However, PD-L1 expression is also upregulated by variety of tumorigenesis-related factors, including: EGFR in NSCLC [113]; the oncogenic BRAF V600E mutant in colorectal cancer [114]; or the loss of PTEN and activation of the PI3K pathway in glioma [115]. As mentioned above, activation of mitogenic pathways (e.g., EGFR, BRAF, or PI3K) entail modifications of the cancer cell milieu, making it more prone to OV infection.

Tumor-induced defects to IFN signaling form a class of mechanisms for altering the interactions of immune cells and malignant cells, with unique implications for oncolytic virotherapy. The uniqueness of such defects stems from the breadth of the IFN response that concomitantly regulates hundreds of immune-mediators [1], many of which directly inhibit different stages of viral infection. In light of the multiple steps involved in the induction, signal transduction and cellular response to IFNs, cancer-induced defects to IFN signaling occur through a plethora of molecular mechanisms including: (i) perturbations to the expression of the IFN receptor; e.g., the ubiquitination and downregulation of the type I IFN receptor (IFNAR1) following inflammatory signaling, nutrient deprivation or hypoxia (all conditions prevalent in the TME) [116,117]. Such down regulation, which was observed in melanoma and colorectal cancer [118,119], is associated with increased metastatic propensity and with the generation of an immune-privileged TME; (ii) perturbations to JAK/STAT1 signaling including epigenetic silencing and inactivating mutations in JAK1 [120,121,122]. In this context, whole-exome and RNA sequencing, and reverse-phase protein array data from different the Cancer Genome Atlas (TCGA) datasets (skin cutaneous melanoma, breast invasive carcinoma, lung adenocarcinoma, and colorectal adenocarcinoma) revealed alterations in *JAK1* or *JAK2* in 5–12 % of the samples, with dependence on cancer type [123]; (iii) crosstalk of JAK/STAT1 signaling with pro-tumorigenic signaling pathways; such as the inhibition of IFN-induced expression of inflammatory genes following STAT3 activation [124].

An interesting aspect of the interactions between immune and malignant cells pertains to the identity (source) of cancer-cell derived immune stimuli. In this context, viruses cause ~15 percent of cancer cases [125], and may thus supply PAMPs for immune-stimulation in virus-transformed cancer cells. However, the majority of tumors do not necessarily encounter pathogens in the course of their developments. A major additional source of stimuli are mutations, which are recognized as tumor-associated antigens and play a prominent immunostimulatory role [126]. Additionally, damage (or danger) associated molecular patterns (DAMPs), which activate PRRs, may also contribute immune-activating stimuli. Thus, DNA fragments generated as a result of genomic instability [127] or upon therapeutic induction of double-stranded DNA breaks [128], activate cGAS/IFN-mediated responses [129], serving thus as a source of immunostimulatory cytokines. Similarly, cytoplasmic exposure of mtDNA [130], resulting from inhibition of the tumor suppressor ataxia telangiectasia mutated (ATM) protein, entails PRR-mediated activation of type I IFN responses [131]. These scenarios support the notion that PRR-mediated activation of type I-IFN responses occurs throughout tumorigenesis, and may force the cancer cell to hamper such responses in order to escape the anti-proliferative and the immune-stimulatory effects of IFN signaling. As mentioned above, such hampered responses optimize the cancer cell milieu towards OV replication.

### 3.2. Acquired Resistance to Immunotherapy, An Additional Source of Modifications to Tumors Which Can Be Exploited by OVs

Acquired resistance to immunotherapy can be viewed as an acute case of tumor immunoediting. In the context of immunotherapy, the release from the constraints imposed by the immune checkpoints, enforces high selective pressure applied on cancer cells by TME-localized immune cells. Thus, clustered regularly interspaced short palindromic repeat (CRISPR)/CRISPR associated protein 9 (Cas9)—mediated knockout screens identified genes related to IFN-γ, in addition to TNF-α and antigen presentation pathways as required for the T-cell mediated killing and its enhancement by anti-PD1 antibodies [132,133,134]. Similarly, truncation in the β2-microglobulin gene resulting in defects in MHC-I-mediated antigen presentation and loss-of-function mutations to JAK1 or JAK2, implying defects to the transduction of antiviral IFN signals; mediate resistance to PD-1 blockade in melanoma [135]. Given that immunostimulatory roles for PRRs have been identified in immunotherapy settings [136,137,138,139,140], they may also be targeted in acquired resistance to this form of therapy, with profound implications to the susceptibility of such edited tumors to OVs. Together, these studies show how escape from immune pressure, in the context of immunoediting in the course of tumor progression, or in the context of immunotherapy; can directly contribute to reduced resistance to infection of cancer cells with OVs.

## 4. Oncogene-Induced Silencing of Immune Genes by DNA Methylation

Methylation of cytosines within CpG dinucleotides is a highly abundant epigenetic modification of mammalian genomes [141]. Methylation patterns, which regulate gene expression, are dynamically regulated via the opposing activities of enzymes that introduce or remove this modification, known as ‘writers’ and ‘erasers’, respectively. This regulatory apparatus is complemented by chromatin ‘readers’, i.e., protein modules that recognize histone and DNA modifications [142]. In accord with the deregulation of methylation in cancer development, DNA methyl transferases (DNMTs, 1, 3A and 3B) are overexpressed in many tumors [143,144,145,146]. A connection between tumorigenic features of cancer cells, epigenetic silencing and defects in antiviral responses is already observed upon spontaneous immortalization of fibroblasts which results in epigenetic silencing of ISGs [147]. Numerous studies reported on promoter methylation and down regulation of different IRFs (e.g., different combinations of IRF4, IRF5, IRF6, IRF7) in cancers, including fibrosarcoma [148], melanoma [149], lung cancer [150], and gastric cancer [151]. Similarly, the promoter of IFN-γ was shown to be methylated in cervical cancer [104]. Moreover, our analysis of the TCGA skin cutaneous melanoma (SKCM) database revealed significantly higher methylation of promoters of genes presenting highly-correlated expression with STAT1 (a gene group that is enriched for cell autonomous immunity genes), as compared to randomly selected genes [152]. In accord with its tumor-promoting functions, RAS was termed as “silent assassin”, due to its gene silencing abilities in cancer cells [153]. In this context, DNMT1 expression is transcriptionally regulated by RAS-induced signaling pathways [143,154,155]. RAS-mediated transformation also modulates the function of DNA-methylation readers such as MBD2 [156]. Furthermore, the expression of enzymes that revert DNA methylation (ten-eleven translocation (TET) methylcytosine dioxygenases) is also regulated by oncogenic signaling in general, and RAS signaling in particular; and ERK-mediated suppression of TET1 is required for K-RAS-induced cellular transformation and hypermethylation of DNA [157]. In accord with a functional interaction between RAS and DNMTs in mediating pro-tumorigenic features, a genome-wide RNA interference (RNAi) screen in K-RAS-transformed NIH-3T3 cells identified DNMT1 and members of the RAS/MEK pathway (ERK2 and MAP3K9) as required for the silencing of the pro-apoptotic FAS gene [158]. The role of such functional interaction in mediating the suppression of immune responses is observed in the downregulation of the RAS-effector MYC and the upregulation of ISGs in lung cancer, following DNMT inhibition [159]. Interestingly, promoter methylation of IRF7 and enhancement of viral infection was observed in nasal epithelial cells exposed to cigarette smoke [160], suggesting that exposure to carcinogens may already set the stage for the silencing of immune genes observed in malignant cells.

## 5. Naturally Oncolytic Viruses Exploit the Altered Cancer-Cell Milieu

The combined metabolic and defense-defective features of the cancer cell milieu (see schematic depiction in Figure 1) can be exploited in the context of oncolytic virotherapy. This is particularly relevant for viruses that are naturally devoid of human disease-causing potential but retain the potential to replicate in, and kill, malignant cells. Such viruses are referred to here as “naturally oncolytic” viruses, to differentiate them from “armed/engineered oncolytic viruses”. Examples of “naturally oncolytic” viruses include attenuated clones of human pathogens (e.g., vaccine clones of measles and mumps viruses, [161]), viruses of veterinary origin (e.g., Newcastle disease virus (NDV), VSV, rat parvovirus (H-1PV), [162,163,164,165]) or the mammalian reovirus, a virus naturally devoid of disease-causing potential [47]. Indeed, we explored the complete absence of IFN signaling in LNCaP prostate cancer cells [120,121], which also present oncogenic KRAS mutation [166], to select an oncolytic mutant of the epizootic hemorrhagic disease virus (EHDV), an orbivirus (arbovirus of the Reoviridae family) that naturally targets ruminants, and that we named EHDV-Tel Aviv University (EHDV-TAU) [120]. Our studies demonstrate productive infection of EHDV-TAU in cells with defective IFN/antiviral responses, e.g. the absence of JAK1 expression/function in LNCaP prostate cancer cells [120,167], or the low basal expression levels of PRRs and defective induction of IFN (following viral infection) by B16F10 murine melanoma cells [152]. Moreover, in the latter case, treatment with inhibitors of epigenetic silencing restored PRR expression and viral induction of IFN responses in the B16F10 cells; exemplifying the role of epigenetic silencing of IFN/ISGs in the cancer cell, as a mechanism for OV selectivity. Additionally, our studies revealed that while productive infection was inhibited upon treatment with IFN, EHDV-TAU retained its cell killing potential of LNCaP cells engineered to express JAK1 (LNCaP-JAK1), when infection was carried out in presence of interleukin-6 (IL-6), an inflammatory cytokine and strong activator of cell autonomous immunity [167]. Thus, with dependence on the cellular setting, OVs may also exploit antiviral responses for induction of cancer cell death.

## 6. Endogenous Retroviruses, Viral Mimicry That Elicits Anti-Tumor Immunity

Tumor cells often show enhanced DNA methylation at CpG-rich sites, located in endogenous retroelements (reviewed in [168,169]). These elements, which make up more than 40% of the human genome, consist of repetitive sequences that belong to three major classes: endogenous retroviruses (ERVs), short interspersed nuclear elements (SINEs) and long interspersed nuclear elements (LINEs). Endogenous retroelements have originated from ancient infections by exogenous retroviruses, which integrated their genomes into the genome of germ cells of the host. This allowed for the vertical transmission of these elements to the offspring of the infected host. During evolution, the majority of such elements have accumulated excessive DNA mutations that inactivated their genes. However, a minority (thousands) retained some of their protein coding potential. Importantly, peptides that are derived from human endogenous retroviruses (hERVs) can be recognized by immune cells. This is exemplified by the infiltration of T cells with receptors specific for hERVs-derived epitopes, into hERVs-expressing clear cell renal cell carcinoma tumors [170]. Furthermore, endogenous retroelements may express additional immunostimulators since transcription of these elements may generate dsRNA molecules (by bidirectional transcription, as well as by sense–antisense pairing); and if reverse transcription follows, complementary DNA (cDNA) and double-stranded DNA (dsDNA) may be created too. These products, which mimic viral infection, may then be sensed by endosomal TLR3, 7, 8 or 9, and/or by cytoplasmic PRRs, including RIG- I, MDA5, cGAS [168,169]. Sensing this ‘viral mimicry’, activates antiviral signaling cascades, including an IFN response (see [171] and additional examples below). ERVs are repressed by variety of mechanisms, including epigenetic silencing through DNA methylation and histone modifications (reviewed in [172,173,174,175]).

Given the potential immunogenicity of endogenous retroelements and their epigenetic suppression, reactivation of these elements by epigenetic modifiers in cancer cells may results in the abovementioned viral mimicry, leading to an anti-cancerous state. For example, treatment of colorectal or ovarian cancer cells with DNMT inhibitors (DNMTis) results in induction of transcription from otherwise suppressed ERVs, the subsequent formation of dsRNA from specific ERV elements, recognition of these dsRNA molecules by MDA5/TLR3 sensors, activation of the mitochondrial antiviral-signaling protein (MAVS)-IRF7 axis and induction of IFN. Together, these result in enhanced anti-proliferative/apoptotic responses [176,177].

The complex interactions among oncogenic signaling, epigenetics and viral mimicry can be further demonstrated by the effects of the cyclin-dependent kinases 4 and 6 (CDK4/6) on cancer immunity [178]. CDK4/6, which interact with D-type cyclins, are central drivers of the cell cycle at the G1-S transition, transduce variety of mitogenic signals and their activity is associated with oncogenesis of several types of cancer (recently reviewed in [179]). Upon the induction of mitogenic signal, cyclin D-CDK4/6 complex promotes retinoblastoma (Rb) phosphorylation, leading to the release of transcription factor E2F from the Rb-E2F complex, and entry into S phase and DNA replication. One of the many targets of E2F is the *Dnmt1* gene [178,180]. Accordingly, CDK4/6 inhibition reduces DNMT1 activity, which leads to activation of ERVs expression, formation of ERVs dsRNA and IFN responses to this viral mimicry. Overall, this increases tumor antigen presentation and, together with additional effects of the CDK4/6 inhibitors, leading to cytotoxic T-lymphocytes (CTL)-mediated clearance of the tumor cells in mouse models [178]. Thus, mitogenic signals suppress ERVs expression via DNA methylation, mediated by the CDK4/6-Cyclin D-Rb-DNMT1 axis, and inhibition of this axis results in ERVs activation followed by enhanced anti-tumor immunity.

## 7. Concluding Remarks

IFNs and ISGs mediate antiviral and tumor-suppressor functions, via cell-autonomous and non-cell autonomous mechanisms. Tumor cells silence IFNs and ISGs along tumorigenesis, and in pronounced fashion in the context of immunoediting. OVs exploit the IFN/ISG-silenced cellular context for replication, and exert part of their therapeutic benefit through stimulation of anti-tumor immunity. Similar to what is observed in OV-infected cells, reversal of DNA methylation-mediated epigenetic silencing of hERVs stimulates anti-tumor immunity through viral mimicry. While the possibility OV/DNMTi combinations may be attractive due to their immunostimulatory potential, the activation of cell autonomous immunity by DNMTi is predicted to be inhibitory towards viral replication. Indeed, our studies showed inhibition of productive infection of EHDV-TAU and oncolytic VSV following DNMTi treatment of murine melanoma cells. However, while the cell-killing potential of oncolytic VSV was diminished in presence of DNMTi, EHDV-TAU retained its cell-killing potential under these conditions ([152], see schematic depiction in Figure 3). This difference in outcome of combined OV/DNMTi treatment, supports the notion of tailoring therapy combinations to the distinct proprieties of different OVs.

## Figures and Tables

**Figure 1 cancers-13-00939-f001:**
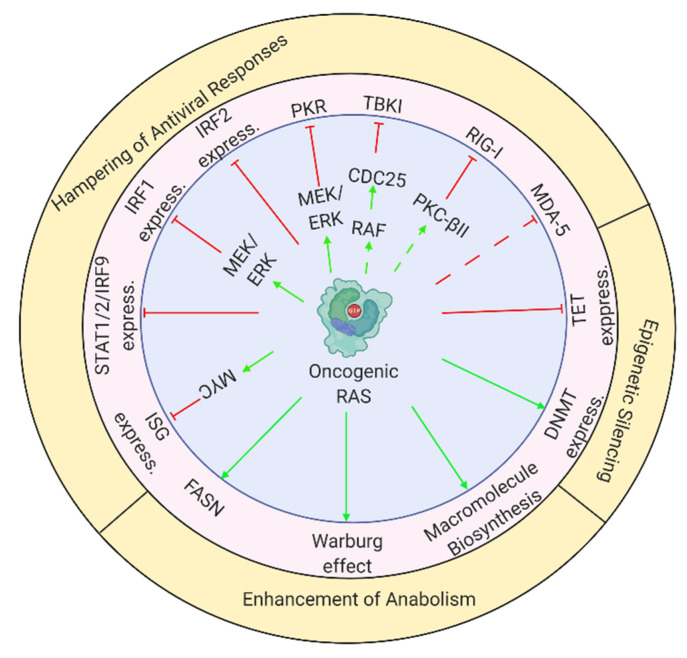
Oncogenic RAS supports viral infection through multiple molecular mechanisms. Scheme depicts mechanisms described throughout review. Green arrows or blunt red arrows denote stimulation or inhibition, respectively. Dashed arrows indicate cases where one source of information supports the connection between oncogenic RAS and its effector, and another source supports the link between the effector and the oncolysis-regulating mechanism. The figure was created with BioRender.com (accessed on 12 February 2021).

**Figure 2 cancers-13-00939-f002:**
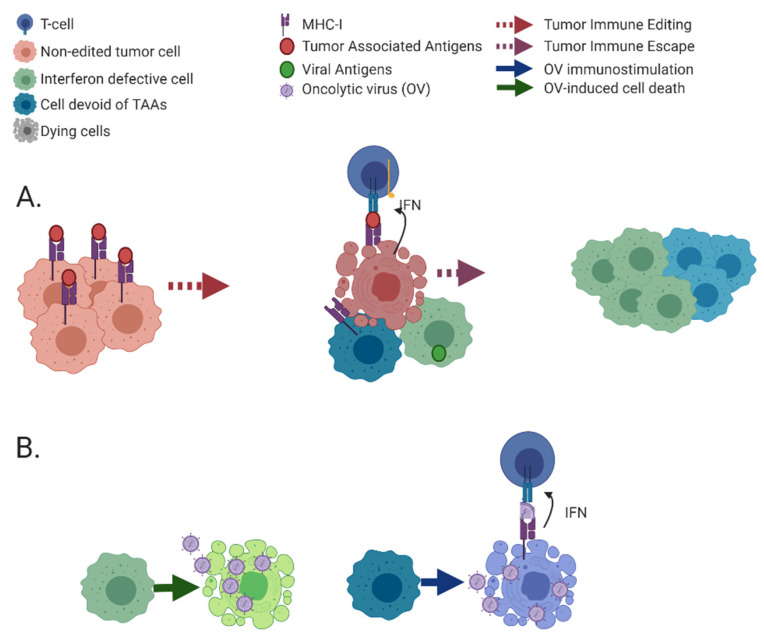
Tumor immunoediting and treatment of escape mutants with oncolytic viruses. (**A**) Tumor cells prior to editing are depicted (in pink) at the left side. Anti-tumor immunity kills a portion of susceptible tumor cells while selecting for escape mutants (middle), allowing their subsequent clonal expansion (right). Two types of escape mutants are depicted: green—IFN-defective cells, blue—cells devoid of tumor-associated antigens. (**B**) OV treatments (e.g., by naturally oncolytic viruses, see Section 5 for definition) of the immunoedited tumors (described in A). Direct cell killing by OVs (left), immune-mediated killing of infected cells (right). A number of such naturally oncolytic viruses are now under clinical trials for treatment of diverse cancer types. The figure was created with BioRender.com (accessed on 12 February 2021).

**Figure 3 cancers-13-00939-f003:**
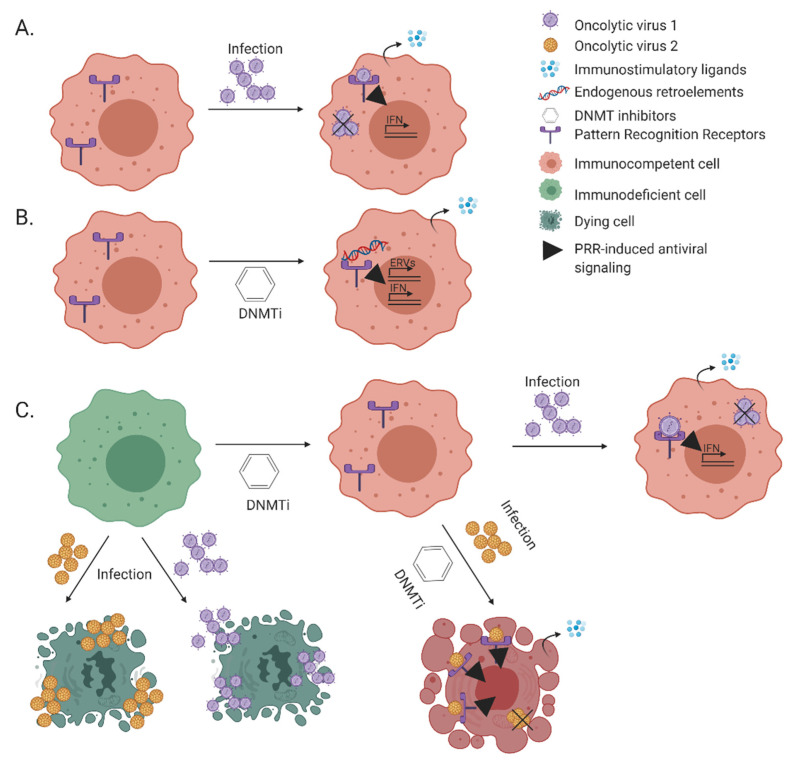
Immuno-stimulation: the interplay between OVs and DNMTis. (**A**)—Infection of oncolytic virus is aborted in an immunocompetent tumor cell in a process involving PRR-mediated PAMP sensing and secretion of immunostimulatory ligands, e.g., interferons. (**B**)—Immunocompetent tumor cells, when treated with DNMTis, may express endogenous retroelements and sense their products. This process is termed viral mimicry. The secreted immunostimulatory cytokines (in A or B) may act in autocrine or paracrine fashions, to induce anti-proliferative/cell death effects at the level of the cancer cell, and/or immune-mediated anti-tumorigenic effects (not depicted for sake of simplicity). (**C**)—The subset of immunodeficient cancer cells (green cells) can be killed directly by replicating oncolytic virus. DNMTi treatment renders them immunocompetent (green to pink shift). When combined with oncolytic viruses, the outcome of such treatment depends on the identity of the oncolytic virus. In a subset of cases (e.g. with EHDV-TAU, yellow viruses) the combined treatment induces cell death, in spite of the DNMTi-mediated reduction in viral replication (as in [152]). The figure was created with BioRender.com (accessed on 12 February 2021).

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
