# Peer review of "Oncolytic Virotherapy: The Cancer Cell Side"

_cancers, 2021, doi:10.3390/cancers13050939_

Round 1

Reviewer 1 Report

The review authored by Ehrlich et al. aims at describe the usage of oncolytic viruses, epigenetic modifiers and ICIs as strategies to activate the anti-tumor immunity.  In my opinion it is difficult to follow the text and the chapters are not very well connected between each other.

Major revision

  • I would have expected to find a dedicated paragraph about oncolytic adenoviruses and their usage as immunosensitizing agents (Ranki T, J Immunother of Cancer 2016, Kuryk L Cancers 2020). A specific section concerning these aspects have to be included.
  • Resistance to immunotherapy has been found in clinical trials. Authors should report the reasons for the limited efficacy and survival
  • Also a clear explanation of ICIs and their mode of action is missing

Author Response

Principal changes to the manuscript:

  1. (i) Alterations to the text so as to sharpen the focus on the cancer cell and its interactions with oncolytic viruses.

(ii) We have divided the text into numbered sections for easier understanding.

(iii) We specifically explore two sources of changes to the cancer cell:

  • Intrinsic- exemplified throughout the review as alterations stemming from oncogenic RAS (Section 2).
  • Extrinsic- exemplified by the tumor-editing process (Section 3).

We also describe DNA methylation as a potential mechanism by which the oncogene alters the cell autonomous immune response (Section 4), and how reversal of this silencing may induce viral mimicry (sensing of endogenous retroelements, Section 6).

We have removed sections referring to immune checkpoint inhibitors, as they are now beyond the scope of the present review. Moreover, we state clearly that the review does not aim to review the differences and specificities of oncolytic viruses in general, as such details require additional reviews, which can be found in this special issue of Cancers.

  1. The revised text aims to be more concise and structured. The number of alterations in text is too large to individually enumerate. However, it should be noted that we have altered the order of the sections, and have added a figure (Figure 1 in the revised text), in which we present the manner by which our chosen model oncogene (oncogenic RAS) alters the cancer cell towards optimization of viral oncolysis.

Specific points:

Reviewer 1

The review authored by Ehrlich et al. aims at describe the usage of oncolytic viruses, epigenetic modifiers and ICIs as strategies to activate the anti-tumor immunity. In my opinion it is difficult to follow the text and the chapters are not very well connected between each other.

As mentioned above, our aim is to focus on the cancer cell. This includes:

  • The manners by which it is optimized towards oncolytic viral replication (inhibition of antiviral responses and metabolic reprogramming).
  • The sources of such changes (intrinsic to the cell - oncogene activity exemplified by RAS; extrinsic to the cell - immunoediting).
  • DNA methylation as a molecular mechanism for oncogene-induced alterations to the above aspects of the cancer cell, and which upon its reversal may unleash viral mimicry, which exposes the cancer cells to many of the stimuli (e.g. PAMPS) stemming from OV infection.

Overall, we appreciate the reviewer's opinion, agree with it and accordingly, generated a majorly revised text. Accordingly, we introduced alterations to the order and contents of the text, added numbering to the sections, and created an additional detailed figure (now Figure 1). These changes should improve following the text.

  • I would have expected to find a dedicated paragraph about oncolytic adenoviruses and their usage as immunosensitizing agents (Ranki T, J Immunother of Cancer 2016, Kuryk L Cancers 2020). A specific section concerning these aspects have to be included.

As explained above, the present review refrains from expanding on specific oncolytic virus types (but rather focus on the cancerous cell). The exception to this is section 5 which mentions naturally oncolytic viruses, as these specifically explore the weaknesses (susceptibilities) of the cancer cell. Specifically, the excellent study: "Phase I study with ONCOS-102 for the treatment of solid tumors - an evaluation of clinical response and exploratory analyses of immune markers. J Immunother Cancer. 2016 Mar 15;4:17. doi: 10.1186/s40425-016-0121-5." deals with an armed virus, and thus, does not fall within this scope. In contrast, we do cite: Cascallo M, Capella G, Mazo A, Alemany R. 2003. Ras-dependent oncolysis with an adenovirus VAI mutant. Cancer Res 63: 5544-50; as it falls within the scope of oncogenic RAS dependency, which is discussed in the review.

  • Resistance to immunotherapy has been found in clinical trials. Authors should report the reasons for the limited efficacy and survival

We fully agree with the reviewer that discussion on the reasons of the failure (limited efficacy) of immunotherapies is a timely and interesting subject. However, this falls beyond the scope of the present review. We specifically refer to acquired resistance to immunotherapy as an additional source of modifications of the tumor cell which optimize it towards OV replication. This is now clarified in the subtitle of the section:

" Acquired resistance to immunotherapy, an additional source of modifications to tumors which can be exploited by OVs."

  • Also a clear explanation of ICIs and their mode of action is missing

Given that we have more accurately focused the review (essentially removing sections referring to ICIs), we consider that such explanation is beyond the scope of the present review.

Reviewer 2 Report

Although the paper presents an interesting topic and is well written with an active voice, I have some major comments and concerns that need to be addressed and clarified. 

1. It would be better to add information about the period covered by this review article. This is to ensure that the paper is up-to-date.

2. Another section named, Methodology would be useful to be added to the paper to provide the readers with information about what type of strategies/methods were used or followed to ensure the quality of processing the collected data and the outcomes. Also, information about the used databases for collecting and or extracting the data. Alternatively, this information could be highlighted in the Introduction section.

3. I recommend the authors add a section named oncolytic viruses and their associated cancers. This section could include the most frequent viruses (RNA and DNA viruses) that infect humans, leading to cause various types of cancer The relation between these viruses and the generated tumors should be added.

4. The paper lacks information about what types of technologies are used in oncolytic virotherapy (for example, CRISPR/Cas genome editing technique) and their applications in cancer research. In my opinion, these points will enhance the quality of the paper.

5. Information presented in Figure 1 is a little bit confusing. It should be highlighted that whether the proposed treatment strategies were validated in preclinical (in vitro or in vivo), clinical studies or not. Please, consider and clarify this point. Similarly, such information should also be applied in Figure 3.

Author Response

Principal changes to the manuscript:

  1. (i) Alterations to the text so as to sharpen the focus on the cancer cell and its interactions with oncolytic viruses.

(ii) We have divided the text into numbered sections for easier understanding.

(iii) We specifically explore two sources of changes to the cancer cell:

  • Intrinsic- exemplified throughout the review as alterations stemming from oncogenic RAS (Section 2).
  • Extrinsic- exemplified by the tumor-editing process (Section 3).

We also describe DNA methylation as a potential mechanism by which the oncogene alters the cell autonomous immune response (Section 4), and how reversal of this silencing may induce viral mimicry (sensing of endogenous retroelements, Section 6).

We have removed sections referring to immune checkpoint inhibitors, as they are now beyond the scope of the present review. Moreover, we state clearly that the review does not aim to review the differences and specificities of oncolytic viruses in general, as such details require additional reviews, which can be found in this special issue of Cancers.

  1. The revised text aims to be more concise and structured. The number of alterations in text is too large to individually enumerate. However, it should be noted that we have altered the order of the sections, and have added a figure (Figure 1 in the revised text), in which we present the manner by which our chosen model oncogene (oncogenic RAS) alters the cancer cell towards optimization of viral oncolysis.

Specific points:

Reviewer 2

Although the paper presents an interesting topic and is well written with an active voice, I have some major comments and concerns that need to be addressed and clarified. 

We thank the reviewer for the overall positive assessment of our review, and hope that the assessment of our revised manuscript will be even more positive.

  1. It would be better to add information about the period covered by this review article. This is to ensure that the paper is up-to-date.

We have chosen not to place a date on the reviewed period as it spans from classical studies (that have contributed essential discoveries to the subject matter), to very recent studies (necessary to being up-to-date). However, we would like to point out that 69 of the references describe manuscripts published from 2015 and later (specifically, 14 from 2020, 12 from 2019, 8 from 2018, 12 from 2017, 11 from 2016, 12 from 2015).

  1. Another section named, Methodologywould be useful to be added to the paper to provide the readers with information about what type of strategies/methods were used or followed to ensure the quality of processing the collected data and the outcomes. Also, information about the used databases for collecting and or extracting the data. Alternatively, this information could be highlighted in the Introduction section.

We thank the reviewer for the suggestion. We specify our methodology in the introduction:

"We begin by focusing on the cancer-cell per-se, analyzing how oncogene-induced modifications serve to optimize the intracellular environment towards OV replication. To this end, we employ RAS-activated pathways as a pivot, exemplifying how this intrinsic oncogenic pathway modulates antiviral responses. We then proceed to focus on the immunoediting of tumors, as this provides a critical extrinsic (selective) source of alterations to cancer-cell autonomous immune functions. Given the overlap in the immune-activation-potential of a cancer cell and its ability to raise antiviral responses, the selective pressure applied by anti-tumor immunity results in both decreased immunogenicity and in defective antiviral responses. We finalize our review by focusing on oncogene-mediated DNA methylation in the context of immune evasion, as an example of how the two processes (oncogenic signaling and immunoediting) converge to influence OVs-cancer-cell interactions, since this epigenetic modification is a prominent molecular mechanism, which silences the cell-autonomous immune responses. In this context, we also discuss the reversal of this form of epigenetic silencing, which may elicit tumor immunogenicity through the expression of endogenous retroelements, thus generating a “viral mimicry” state, emulating the immune-stimulatory potential of OVs."

  1. I recommend the authors add a section named oncolytic viruses and their associated cancers. This section could include the most frequent viruses (RNA and DNA viruses) that infect humans, leading to cause various types of cancerThe relation between these viruses and the generated tumors should be added.

The focus of the present review is on the cancer cell susceptibility to oncolytic viruses. As such, expanding on oncogenic viruses is beyond the focus of the present review.

  1. The paper lacks information about what types of technologies are used in oncolytic virotherapy (for example, CRISPR/Cas genome editing technique) and their applications in cancer research. In my opinion, these points will enhance the quality of the paper.

The focus of the present review is on the susceptibility of cancer cells to oncolytic viruses. A detailed description of the molecular features of the different oncolytic viruses being developed/used is beyond our focus. Moreover, we assume that such analyses are part of the additional reviews, which form this special issue. We specifically refer to this in the text: "As such, this review concentrates on the malignant cell, while detailed description of different OVs can be found in the accompanied reviews of this issue."

  1. Information presented in Figure 1 is a little bit confusing. It should be highlighted that whether the proposed treatment strategies were validated in preclinical (in vitroor in vivo), clinical studies or not. Please, consider and clarify this point. Similarly, such information should also be applied in Figure 3.

We have altered the figures to focus specifically on "naturally oncolytic viruses" (see section 5). We have also added the following text to the figure legend: " OV treatments (naturally oncolytic viruses, see section 5 for definition) of the immunoedited tumors. Direct cell killing by oncolytic viruses (left), immune-mediated killing of infected cells (right). A number of such naturally oncolytic viruses are now under clinical trials for treatment of diverse cancers".

Round 2

Reviewer 1 Report

The authors replied to most of my comments, I suggest the MS for publication

Reviewer 2 Report

The paper has been significantly improved.